# Microstructure and Properties of WC/Ni-Based Laser-Clad Coatings with Different WC Content Values

**DOI:** 10.3390/ma15186309

**Published:** 2022-09-11

**Authors:** Xuehui Shen, Hao Peng, Yunna Xue, Baolin Wang, Guosheng Su, Jian Zhu, Anhai Li

**Affiliations:** 1School of Mechanical Engineering, Qilu University of Technology (Shandong Academy of Sciences), Jinan 250353, China; 2Shandong Institute of Mechanical Design and Research, Jinan 250031, China; 3Key Laboratory of High Efficiency and Clean Mechanical Manufacture of MOE, School of Mechanical Engineering, Shandong University, Jinan 250061, China

**Keywords:** WC/Ni-based composite coating, laser cladding, nanoindentation

## Abstract

The purpose of this work is to investigate the effect of the WC content on the surface characteristics and nanoindentation behaviors of WC/Ni-based composite laser-clad coatings. Four NiCrSiBC coatings with WC wt% of 30%, 40%, 50%, and 60%, respectively, were clad on carbon steel substrates using a laser. The morphologies and phase compositions of four clad coatings were comparatively observed. In addition, the hardness and elastic modulus values of the four coatings were measured and quantitatively calculated. As a result, with the increase in WC, the coating grains were more refined. Meanwhile, cracks and WC particle breakage occurred in the 50–60% WC coatings, whereas this was not found in the 30–40% WC coatings. When the WC content increased from 40% up to 50%, the coating hardness and elastic modulus significantly increased. However, a further increase in WC from 50% to 60% did not result in considerable improvement in coating quality but considerably worsened the coating’s cracking behavior instead. Therefore, for WC/Ni-based composite coatings, a threshold exists for the WC content, and this value was 50% within the experimental scope of this study.

## 1. Introduction

As a popular coating material, nickel-based alloy powders have been widely applied in various industrial fields due to their excellent self-compatibility and wettability [1,2]. In addition, WC particles have a high melting point, high hardness, and good wettability with bonding metals. Therefore, WC particles are often added as reinforcements in Ni-based alloy powders to form ceramic–metal composite coatings [3,4]. Meanwhile, depositing a layer of high-performance coating on a cheap metal matrix saves costs with less environmental consumption [5]. In addition, laser cladding can obtain excellent properties of the cladding material with less influence on the substrate [6]. Farahmand et al. [7] prepared Ni-based/WC coatings via laser cladding and found that the added nano-WC particles improved the hardness and wear resistance of the coatings. Singh et al. [8] deposited a NiCrSiBC + 50WC composite coating via laser cladding and found that high scanning speed, high laser power, and intermediate powder feeding rate could improve the coating’s wear resistance. Cheng et al. [9] fabricated a Ni-based/WC composite coating on NAK80 mode steel and proved that WC phases served to improve wear resistance due to the reinforced phases of Cr_23_C_6_, WC, and W_2_C dispersed in the coating. Zhou et al. [10] reported that crack formation in clad coatings was highly related to the distribution and dissolution of WC particles in Ni-based composite coatings.

In the framework of coating preparation techniques, compared with other common methods such as spraying and physical vapor deposition (PVD), laser cladding technology has unique advantages due to its low dilution rate, small matrix deformation, shallow heat-affected zone, and metallurgical bonding between the coating and the matrix [11]. However, due to the inherent characteristic of rapid heating and solidification of laser cladding, both thermal stress and microstructure stress could be induced inside the coatings during cladding, which causes cracks [12]; this is a more significant issue for hard ceramic–metal materials. 

Many efforts have been made to improve the surface properties and service performance features of ceramic–metal clad coatings, including adjusting the weight percentage of ceramic phases and adding rare earth elements. Wang et al. deposited a gradient composite coating with a four-layer structure on a Q345R steel substrate and proved that a multi-layer structure was beneficial to reduce the coating’s cracking susceptibility [13]. Hu et al. clad Ni-based cladding layers with WC content values of 5%, 7.5%, and 10% on a stainless steel substrate and found that the corrosion rate, friction coefficient, and wear volume of the clad layer first increased and then decreased with the increase in WC [14]. Li et al. clad Ni60 coatings with WC content values of 5%, 10%, 15%, 20%, 25%, 30%, and 35%, respectively, on a QT5007 substrate and found that the 20% WC coating had the best corrosion resistance [15].

As stated above, many researchers have prepared various WC/Ni-based composite coatings using lasers, and most reports focus on adjusting the formulation of clad powders [16]. However, the existing literature mainly studied various composite coatings with the WC content varying from 10 wt% to 40 wt%, while few studies investigated coatings with a WC content value of more than 40%. Meanwhile, research focusing on the mechanical behaviors of WC/Ni-based composite coatings is scarce. It has been established that a high level of WC content facilitates the initiation and propagation of microcracks inside clad layers [17]. Therefore, if WC/Ni-based composite laser-clad coatings were to be applied in the engineering practice, it is necessary to clarify the forming mechanism of composite coatings with high WC content, hence the need for this study.

In the current study, four laser-clad coatings with WC content values of 30%, 40%, 50%, and 60%, respectively, were fabricated and comparatively investigated. Furthermore, the micromechanical behaviors of these four coatings were quantitatively evaluated via a nanoindentation test, and the corresponding results were analyzed.

## 2. Materials and Methods

NiCrSiBC alloy powders were manufactured by Tianjin Zhujin Technology Development Corporation Ltd. NiCrSiBC alloy powders with 30%, 40%, 50%, and 60% WC particles were selected as laser-clad materials, and their chemical composition and powder morphology are shown in Table 1 and Figure 1. Four medium carbon steel (0.45% C wt%) blocks with dimensions of 40 mm× 60 mm× 8 mm were prepared as substrates. Before cladding, the four types of composite powders were put into a drying oven to remove moisture.

Before cladding, substrate surfaces were polished with sandpapers to a roughness value of Ra = 1 μm and then cleaned in an ultrasonic acetone bath (23 kHz, 10 min). The four coatings with different WC content values were clad on the substrate using a laser-clad workstation (LYRF-4000W). The processing parameters set in the integrated control system (LYRF1500) of the workstation were as follows: laser power 1 kW, laser spot diameter 2.5 mm, laser scanning speed 1200 mm/min, lap rate 40%, powder feeding speed 11.2 g/min, and protective gas (Ar_2_) flow rate 5 L/min.

After cladding, the coating surfaces were polished with sandpaper, then corroded with aqua regia, and cleaned with absolute ethanol after corrosion.

A scanning electron microscope (Phenom ProX, Phenom-World, Eindhoven, The Netherlands) and its accompanying energy dispersive analyzer (EDS) were used to observe and measure the surface morphology, metallographic structure, and chemical composition content of the four composite coatings. The hardness values of these coatings were measured with a Vickers hardness tester (HXD-1000TMC, Shanghai Taiming Optical Instrument Co., Ltd., Shanghai, China) under a load of 300 gf and a dwell time of 15 s. Phase analysis of the coatings was carried out via X-ray diffraction (Japanese Science SE, Rigaku Corporation, Japanese) using Cu–Kα radiation at 45 kV and 40 mA.

A nanoindentation test was performed with a nanoindenter (NTH_3_, Bruker Company, Karlsruhe Germany) at room temperature. During the test, a Berkovich indenter was pressed into the coating surface at a loading rate of 15 mN/min to a maximum of 30 mN, and the dwell time of the maximum load was 10 s before unloading. For each coating sample, five tests were performed to avoid random errors.

## 3. Results and Discussion

### 3.1. Surface Characterization

The four images in Figure 2 show the surface morphologies of the four composite coatings with WC content values of 30%, 40%, 50%, and 60%, respectively. The EDS measurements of the marked points in Figure 2 are listed in Table 2.

From Figure 2a, it is observed that no visible crack or defect was found in the 30% WC coating surface. According to the EDS measurement, the large angular white particles in Figure 2 were in the WC phase (points A, C, F, and H). In the 30% WC coating, WC particles were sparsely distributed and remained complete. As shown in Figure 2b, for the 40% WC coating, no obvious crack was spotted. However, a certain number of tiny bar-shaped carbides were formed and scattered in the 40% WC coating. This is a normal phenomenon of clad coatings in the laser cladding process [18]. During laser cladding, materials are subjected to a high temperature above 1000 Celsius degree, and some WC particles are thermally decomposed into C and W elements following Equations (1) and (2) [19,20]. These two elements then react with other elements in a molten pool to form differently shaped carbides during the cooling stage [21].
(1)$$2\mathrm{WC}\to W_{2}C+C$$
(2)$$W_{2}C\to 2W+C$$

When the WC content increased up to 50%, as shown in Figure 2c, the amount of bar-shaped carbides noticeably rose. In the carbide-clustered area (point E), the W content reached about 14%; that is, the precipitation of WC particles became more and more serious with the increase in the WC content. Another observation was that several cracks occurred in the 50% WC coating. The root reason for crack formation is the difference in thermophysical properties between WC particles and the binder matrix [22]. In particular, the thermal expansion coefficient of WC is 6.5–7.4 × 10^−6^K^−1^, while this value for NiCrSiBC is 13.3–16.8 × 10^−6^K^−1^. Meanwhile, during the cladding process, coating materials were heated and then cooled in a very short time. Therefore, in the solidification process of coating materials, the shrinkage of the binder matrix was much more significant than that of the WC particles, and therefore, WC particles were subjected to tensile stress, resulting in microcrack initiation [23]. Then, microcracks nucleated at stress concentrators, which were the WC particles, and propagated along the maximum stress direction. Meanwhile, the coatings with high content of WC were more susceptible to cracks. On the one hand, more WC particles meant more tensile stress induced inside the coating. On the other hand, in the hardening phase, more WC particles led to higher hardness but poorer ductility. In addition, the precipitated carbides have been proven to increase tensile stress and, therefore, increase crack sensitivity [10]. Thus, once tensile stress was greater than the fracture strength of the material, cracks occurred. As can be seen in Figure 2d, when the WC content reached 60%, more cracks were formed than that in the case of 50% WC. Furthermore, a few WC particles were seriously damaged besides cracking due to the further increased tensile stress coupled with the reduced fracture strength resulting from a further increase in WC particles. Cracks were more likely to occur due to greater stress in the WC with larger particle sizes [24].

### 3.2. Phase Analysis

Figure 3a shows the XRD patterns of the four coatings. According to their measurements, the main phases of the four coatings were γ-(Ni, Fe), WC, Ni_4_B_3_, M_6_C, M_12_C, M_23_C_6,_ and W_2_C (M = W, Ni, Cr, Fe). Among them, W_2_C was formed from the in situ decomposition of WC particles, which could boost solid solution strengthening and facilitate microstructure stabilization [25]. It can be found from Figure 3b that the peaks of the four coatings had a tendency to shift to higher angles with the increase in the WC content. According to the Bragg equation, peak shifting is considered to be closely related to residual stress induction. The right shift in a diffraction peak indicates that the diffraction angle increases, the inter-planar spacing decreases, and the microstrain increases [26]. In addition, with the WC content increasing from 30% up to 60%, the main peak weakened. Figure 3c shows the full width at the half maximum (FWHM) values for the four coatings. According to Sherrer’s formula, the widening of the main peak means a decrease in grain size [27]. Therefore, it could be analyzed that the grain size of the coating decreased with the increase in the WC content; thus, WC particles are beneficial to grain refinement [19].

The surface metallographic structures of the four tested coatings are shown in Figure 4. Similar observations can be made in Figure 2 and Figure 4. One of the aspects observed is that clear cracks were found in the 50% and 60% coatings, while no cracks were spotted in the other two coatings. Meanwhile, cracking was more severe for the 60% WC coating than for the 50% WC coating. Another observation reveals that no visible carbide was produced in the 30% WC coating, whereas varying amounts of white bar-shaped carbides were precipitated in the other three cases. Furthermore, the number of carbides had a clear growing trend with the increase in the WC content. In addition, when comparing the four images, the grain size showed an apparent decreasing tendency with the increase in the WC content, which was consistent with the XRD results. During the solidification of the molten pool, unmelted WC particles and precipitated carbides acted as nucleation sites and hindered the grains from continuous growth. Therefore, more WC content facilitated grain refinement. It is evident from Figure 4 that the grain size of the 60% WC coating was the least among the four tested coatings.

### 3.3. Hardness

Figure 5a shows the surface microhardness measurements of the four coatings. The surface microhardness of 30–60% WC coatings were 485.2 HV0.3, 528.7 HV0.3, 654.3 HV0.3, and 674.3 HV0.3, respectively. The hardness of the substrate was 243.57 HV0.3. Therefore, the surface hardness values of the four coatings were 1.99, 2.17, 2.69, and 2.77 times that of the substrate, respectively, showing a clear increasing trend with the increase in the WC content. The increase in hardness resulted from the combination of solid solution strengthening and dispersion strengthening. On the one hand, during laser cladding, at a high temperature, WC particles were easy to decompose and precipitate W and C elements. W and C elements then dissolved into a Ni-based alloy solvent during rapid solidification to form supersaturated solutions, leading to the formation of carbides and W-rich reinforcing phases and thus achieving solid solution strengthening. On the other hand, WC particles and the precipitated carbides could serve as nucleation sites and change the growth direction of dendrites [25,28]. A large number of irregular nucleation sites could hinder the growth of dendrites and refine grains, achieving dispersion strengthening. Obviously, more WC particles meant more hardening phases and nucleation sites, and therefore more significant surface strengthening effect; that is, more WC content meant higher coating hardness. However, hardness and toughness are usually two contradictory parameters; an increase in hardness is often obtained at the cost of sacrificing toughness. Therefore, from another side, more WC content could also increase the cracking susceptibility of composite coatings [13].

The cross-section hardness variations in the four coatings are shown in Figure 5b. In these four coatings, in the range of 0–400 μm beneath the top surface, the hardness values remained nearly stable. In the heat-affected zone (HAZ) (400–720 μm beneath the top surface), the hardness values gradually reduced to that of the substrate. Using the two images in Figure 5, it is worth noting that there was a significant hardness gain when the WC content rose from 40% up to 50%, while a slight gain was achieved when the WC content further rose to 60%. Accordingly, it could be analyzed that when the WC content reached a certain threshold, only increasing the amount of WC would not lead to the anticipated hardness gain.

### 3.4. Nanoindentation Behavior

Nanoindentation provides a convenient way to investigate the mechanical properties of materials at the nanoscale level. Figure 6a illustrates the principle of the nanoindentation test. In this principle, hmax represents the maximum penetration depth of the indenter. Once the loading force is removed, the material recovers from partial deformation, and the indentation depth decreases. The final indentation depth is shown as hr, and hc is the elastic recovery depth.

Figure 6b shows the load curves of the four coatings during the loading and unloading processes. At a normal load of 30 mN, the maximum penetration depth (hmax) values of the four tested coatings were 492.94 nm, 469.73 nm, 427.35 nm, and 415.29 nm, respectively, showing a decreasing tendency with the increase in the WC content. That is to say, the WC phase was beneficial to the coating’s deformation resistance.

Figure 6c shows the nanohardness variations in terms of penetration depth. With the applied normal load increasing to a maximum value of 30 mN, in the four cases, the nanohardness values rapidly increased to the maximum, then fluctuated for a short time, and finally stabilized. Fluctuation in hardness is mainly related to the surface roughness of the coating. At the very beginning of nanoindentation, the indenter is only in contact with a single surface asperity, and the asperity plastically deforms under the load. As the indenter goes deeper, other asperities gradually come into contact with the indenter [29]. Therefore, before the penetration depth reached about 100 nm, nanohardness fluctuated. After the indenter went deeper than 100 nm, the contact surface between the coating surface and indenter became flattened, and therefore, nanohardness stabilized. Furthermore, it can be seen from Figure 6c that with the increase in WC, the maximum value of hardness showed an apparent increase, and such an increase was more significant for the 50% and 60% WC coatings. More WC content meant a larger amount of hard phases, including the WC particles and precipitated carbides involved in a unit volume of the coating. Therefore, a larger load was required to deform asperities in coatings with high WC content.

Figure 6d shows the calculated elastic modulus and average nanohardness. Obviously, with the increase in the WC content, the elastic modulus values of the coatings showed a clear growing trend. The elastic modulus values of the 40% WC, 50% WC, and 60% WC coatings were 229.44 GPa, 286 GPa, and 304.21 GPa, which were 8.4%, 35.1%, and 43.6% higher than that of the 30% WC coating (211.69 GPa). It is worth noting that the elastic modulus of the 50% WC coating was 24.7% higher than that of the 40% WC coating, whereas this value for the 60% WC coating was only 6.4% higher than that of the 50% WC coating.

The calculated average nanohardness values of the 30–60% WC coatings were 3.99 GPa, 4.36 GPa, 5.16 GPa, and 5.42 GPa, respectively. From Figure 6d, it can be seen that with the WC content growing from 30% to 40%, nanohardness reached a 9.3% gain, and this growth rate was 18.3% and 5%, respectively, when the WC content increased from 40% up to 50% and then further increased to 60% from 50%. This phenomenon was similar to that shown in Figure 5; that is, when the WC content increased from 40% up to 50%, the coating significantly improved in terms of its elastic modulus and nanohardness, whereas only a slight level of growth was obtained when the WC content further increased. As stated above, with the increase in the WC content, the number of reinforcement phases increased, which served to hinder dislocation and thus refine the grains. Therefore, good nanoindentation behavior of the 50% WC coating was obtained. However, in the case of the 60% WC coating, the molten pool reached a close-to-saturation state, and not many more carbides were precipitated than those in the 50% WC coating, and thus, the desired improvement was not achieved in the nanoindentation behavior.

## 4. Conclusions

Four Ni-base composition coatings with respective WC content values of 30%, 40%, 50%, and 60% were prepared via laser cladding. The surface characterizations and nanoindentation behaviors of these four coatings were compared and analyzed.

(1) For the 30% WC coating, WC particles remained complete with invisible decomposition and carbide precipitation. With the WC content growing from 40% up to 60%, the decomposition of WC particles became more and more severe, and more and more white bar-shaped carbides were precipitated. In addition, for the 50% and 60% WC coatings, cracks occurred. Furthermore, in the 60% WC coating, obvious WC particle breakage was found besides cracks. Crack formation in high WC content cases was attributed to the high tensile stress coupled with low coating fracture strength resulting from a large number of WC particles.

(2) Both the metallographic structures and XRD results showed that the grain size of the coating decreased with the increase in the WC content. During the solidification of the molten pool, WC particles, together with the precipitated carbides, acted as nucleation sites and hindered the grains from continuous growth, and therefore, the grains were refined.

(3) Consistent results were drawn from the microhardness test and nanoindentation test. Specifically, the elastic modulus and coating hardness had increasing trends on the whole with the increase in the WC content. The mechanical property improvement was attributed to solid solution strengthening, dispersion strengthening, and grain refinement. When the WC content grew from 40% up to 50%, both the elastic and hardness behavior significantly improved. However, such improvement was much less when the WC content further increased to 60%, which might have resulted from the saturation of WC particles. As a result, within the experimental scope of this study, when the WC content reached a threshold, a further increase in WC particles did not lead to considerable improvement in the coating’s mechanical properties but rather considerably worsened the cracking. Therefore, for WC/Ni-based composite coatings, a threshold exists for the WC content, and a further increase in WC beyond this threshold does not result in considerable improvement in coating quality but instead significantly worsens the coating’s cracking behavior. Within the experimental scope of this study, 50% was the threshold for the WC content.

## Figures and Tables

**Figure 1 materials-15-06309-f001:**
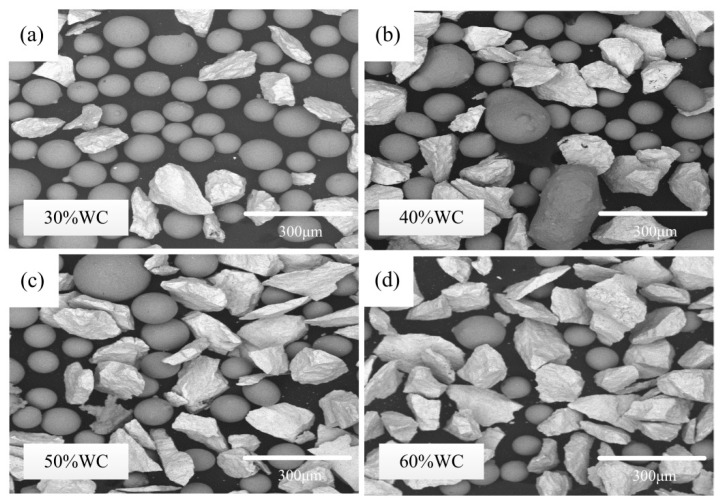
Powder morphologies: (**a**) 30% WC, (**b**) 40% WC, (**c**) 50% WC, and (**d**) 60% WC.

**Figure 2 materials-15-06309-f002:**
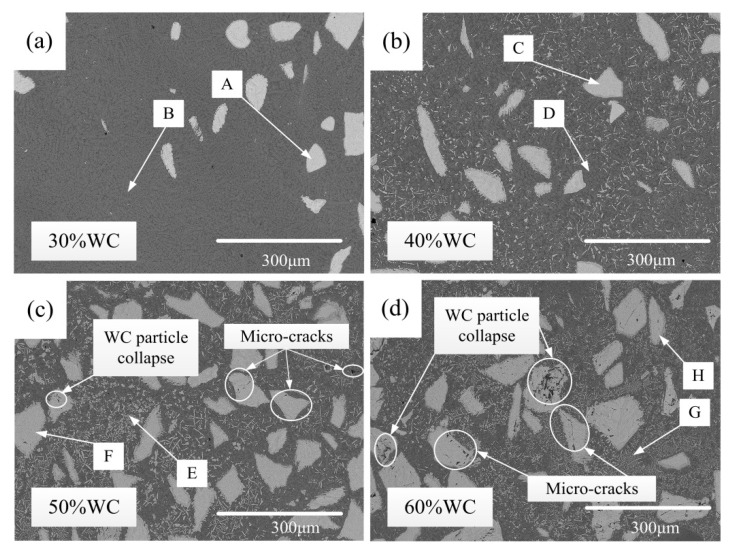
Surface morphologies: (**a**) 30% WC, (**b**) 40% WC, (**c**) 50% WC, and (**d**) 60% WC.

**Figure 3 materials-15-06309-f003:**
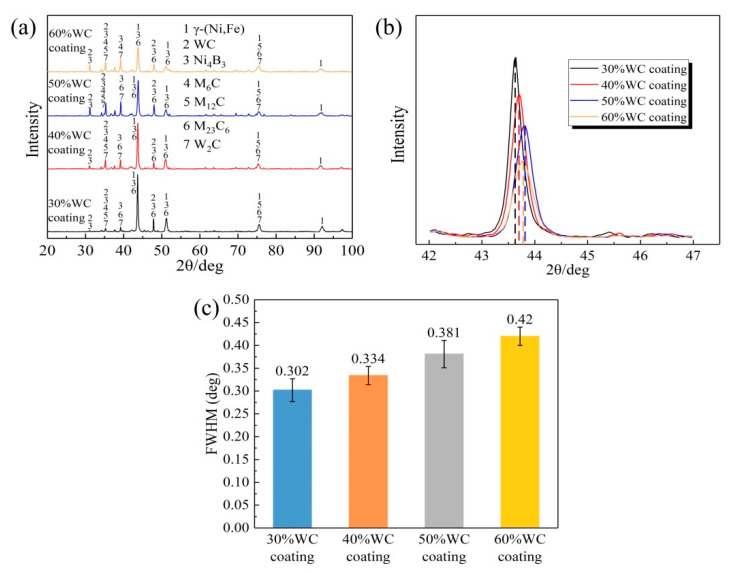
(**a**) XRD analysis of 30–60% WC coatings surfaces, (**b**) the offset of diffraction peak, and (**c**) the FWHM values of the four coatings.

**Figure 4 materials-15-06309-f004:**
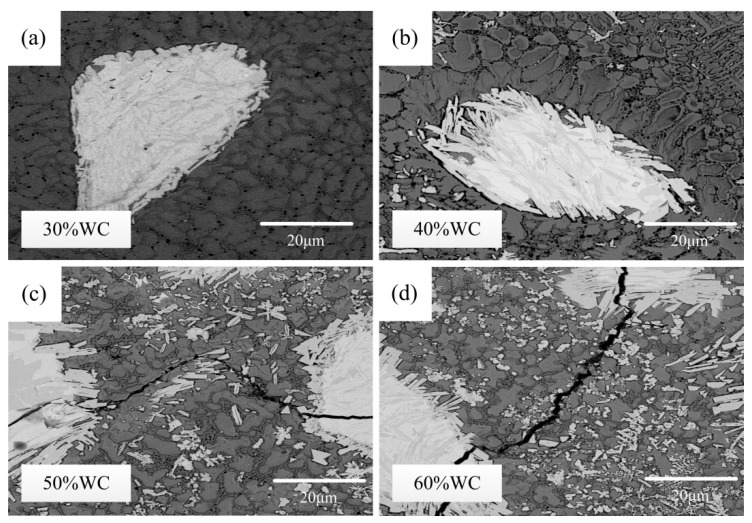
Surface microstructure of samples: (**a**–**d**) 30–60% WC coatings.

**Figure 5 materials-15-06309-f005:**
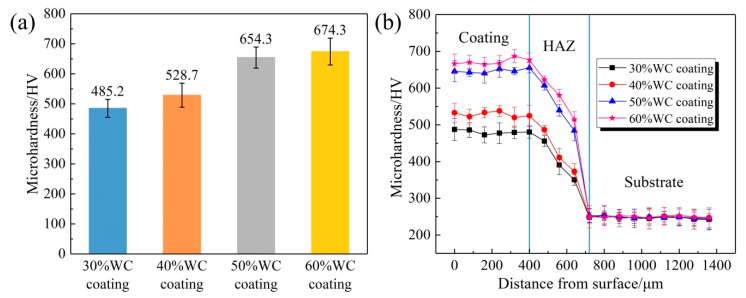
(**a**) The surface hardness of the samples and (**b**) the cross-sectional hardness of the samples.

**Figure 6 materials-15-06309-f006:**
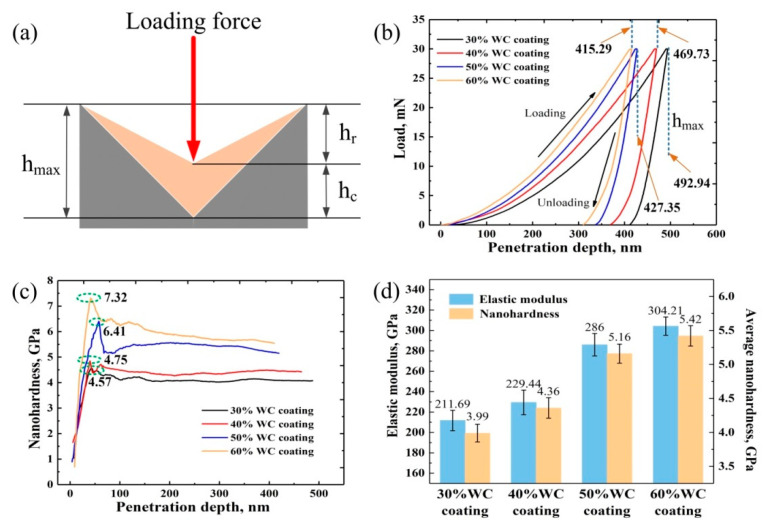
(**a**) The principle of the nanoindentation test, (**b**) nanoindentation p–h curves of 30–60% WC coatings, (**c**) the variations of nanohardness against penetration depth, and (**d**) elastic modulus and nanohardness.

**Table 1 materials-15-06309-t001:** Chemical composition of four powders.

Label	Chemical Composition (wt %)						
	WC	C	Fe	Cr	Si	B	Ni
30% WC	30	0.067	0.043	1.82	1.7	0.71	Bal
40% WC	40	0.05	0.35	1.68	1.6	0.63	Bal
50% WC	50	0.046	0.28	1.55	1.45	0.58	Bal
60% WC	60	0.035	0.21	1.46	1.25	0.45	Bal

**Table 2 materials-15-06309-t002:** EDS results of marker point in Figure 1 (wt%).

Position	Fe	C	Ni	Cr	W
A	0.49	5.23	4.39	-	89.89
B	2.20	2.90	88.35	1.64	-
C	-	5.93	0.82	-	93.25
D	3.53	4.37	88.14	0.9	-
E	3.51	3.38	76.11	0.6	13.79
F	0.42	5.21	-	-	94.37
G	5.09	4.65	73.77	0.72	13.93
H	0.20	6.55	0.60	-	92.65

## Data Availability

The data presented in this study are available from the corresponding author upon reasonable request.

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
