# Peer review of "Microstructure and Properties of WC/Ni-Based Laser-Clad Coatings with Different WC Content Values"

_materials, 2022, doi:10.3390/ma15186309_

Round 1

Reviewer 1 Report

the research is interesting, I request the answer to my comments before finalizing the article.

1- Please indicate in the text the treatment duration in ultrasonic bath and the operation frequency of this equipment.

2- Please correct the numbers presentation in line 130.

3-Why do you not raise the WC rate to 80% in order to increase the micro-hardness.

Reviewer 2 Report

This work deals with laser-clad coatings obtained from a mixture of WC and NiCrSiBC powders. The effect of WC content on the characteristics of the coatings has been studied.

I have several suggestions to improve the paper.

1. Please revise the title of the paper. In the present version, it is not read smoothly. My suggestion is "Microstructure  and properties of WC/Ni-based laser-clad coatings with different WC contents".

2. Not "cladding" coatings but "clad" coatings.

3. Coatings were not "cladded" but "clad".

4. Line 40: correct the chemical formula of chromium carbide.

5. The manufacturer information of the powders and equipment used in the present work should be provided.

6. How was the particle size of WC selected? The optimal content of WC has been found for the particular powder taken for the present study. Is it possible to use a finer powder of WC? How would the results change if finer WC particles were used?

7. English language revision of the text is needed.

Reviewer 3 Report

The authors have investigated the effect of WC content on surface characteristics and nano-indentation behaviors of WC/Ni-based composite laser cladding coatings. The authors used the laser to clad on carbon steel substrates four NiCrSiBC coatings having different percentages of weight % of WC; namely, 30%, 40%, 50%, and 60., I recommend that the authors revise this article to be improved considering the following suggestions;

  • English language for the whole text of the manuscript must be carefully revised.
  • Introduction: From line 34 to line 43, the references from 5 to 8, the authors should cite the reference number directly after the authors’ names, for example; Farahmand et al. [5], etc.
  • Materials and methods: for the images displayed in Fig. 1 and the data shown in Table 1; the authors have not reported how they mixed all the powders together and only mentioned the removal of moisture. How do the authors confirm the homogenous mixing of the elements of the fabricated coatings?
  • Results and discussion: Figure 3 should be replaced with a better-resolution one.
  • References: The authors have cited 21 references; this number must be increased by citing more recent references. Also, the authors have not cited any references belonging to this journal “Materials”, although there are many published articles in this area; this should be considered.
